# Plasma Oxidative Status in Preterm Infants Receiving LCPUFA Supplementation: A Pilot Study

**DOI:** 10.3390/nu12010122

**Published:** 2020-01-01

**Authors:** David Ramiro-Cortijo, Ángel Luis López de Pablo, Mᵃ Rosario López-Giménez, Camilia R. Martin, Joanne Brown, Miguel Saenz de Pipaón, Silvia M. Arribas

**Affiliations:** 1Department of Physiology, Faculty of Medicine, Universidad Autónoma de Madrid, C/ Arzobispo Morcillo, 2, 28029 Madrid, Spain; dramiro@bidmc.harvard.edu (D.R.-C.);; 2Department of Medicine, Beth Israel Deaconess Medical Center, Harvard Medical School, 330 Brookline Avenue, Boston, MA 02215, USA; jbrown17@bidmc.harvard.edu; 3Department of Epidemiology, Public Health and Microbiology, Faculty of Medicine, Universidad Autónoma de Madrid, C/ Arzobispo Morcillo, 2, 28029 Madrid, Spain; mrosario.lopez@uam.es; 4Department of Neonatology, Beth Israel Deaconess Medical Center, Harvard Medical School, 330 Brookline Avenue, Boston, MA 02215, USA; cmartin1@bidmc.harvard.edu; 5Carlos III Health Institute, Maternal and Child Health and Development Research Network, 28046 Madrid, Spain; 6Department of Neonatology, Hospital La Paz, Paseo de la Castellana, 261, 28046 Madrid, Spain

**Keywords:** preterm, supplementation, arachidonic acid, docosahexaenoic acid, antioxidants, oxidative stress

## Abstract

After birth, preterm infants are deficient in arachidonic acid (ARA), docosahexaenoic acid (DHA), and antioxidants, increasing their risk of oxidative stress-related pathologies. The principal aim was to evaluate if supplementation with long-chain polyunsaturated fatty acids (LCPUFAs) improves antioxidant defenses. In total, 21 preterm infants were supplemented with ARA and DHA in a 2:1 ratio (ARA:DHA-S) or with medium-chain triglycerides (MCT-S). Plasma *n*-3 and *n*-6 LCPUFAs were measured at birth, postnatal day 28, and 36 weeks of postmenstrual age (36 WPA) by gas chromatography–mass spectroscopy. Plasma antioxidants (glutathione (GSH), catalase, and thiols) and oxidative damage biomarkers (malondialdehyde (MDA), carbonyls) were analyzed at the same time points by spectrophotometry, and scores of antioxidant status (Antiox-S) and oxidative damage (Proxy-S) were calculated. At 36 WPA, linoleic acid (LA) and dihomo-γ-linolenic acid (DGLA) were decreased in ARA:DHA-S compared to the MCT-S group (LA: ARA:DHA-S = 18.54 ± 1.68, MCT-S = 22.80 ± 1.41; *p* = 0.018; DGLA: ARA:DHA-S = 1.68 ± 0.38, MCT-S = 2.32 ± 0.58; *p* = 0.018). Furthermore, α-linolenic acid (ALA) was increased in ARA:DHA-S (ARA:DHA-S = 0.52 ± 0.33, MCT-S = 0.22 ± 0.10; *p* = 0.018). Additionally, LA:DHA ratio was decreased in the ARA:DHA-S compared to control group (ARA:DHA-S = 6.26 ± 2.35, MCT-S = 8.21 ± 2.65; *p* = 0.045). By the end of supplementation (36 WPA), catalase, thiol groups, and Antiox-S were significantly higher in neonates receiving ARA:DHA-S compared to those receiving MCT-S, with no differences in oxidative stress biomarkers. In conclusion, ARA:DHA supplementation in preterm neonates resulted in an overall improvement in antioxidant to oxidant balance and a decrease in early fatty acid precursors of the n-6 relative to the n-3 pathway. These effects may reduce oxidative stress and inflammation.

## 1. Introduction

Common morbidities diagnosed during neonatal intensive care hospitalization in preterm infants, such as bronchopulmonary dysplasia (BPD), necrotizing enterocolitis (NEC), persistent ductus arteriosus, and retinopathy of prematurity (ROP), are associated with oxidative damage [1,2]. The increased likelihood of oxidative damage is partly related to the immature antioxidant systems of the preterm infant, and it is aggravated by medical interventions such as the use of positive pressure ventilation which increases reactive oxygen species (ROS) generation [3,4,5]. 

Arachidonic acid (ARA; 20:4 *n*-6) and docosahexaenoic acid (DHA; 22:6 *n*-3) are long-chain polyunsaturated fatty acids (LCPUFAs) of the n-3 and n-6 series that are critical for fetal growth and development. During intrauterine life, the fetus acquires DHA and ARA through placental transfer with an ARA:DHA ratio around 2:1 [6]. In preterm infants, this supply is interrupted, and they depend on breastmilk or supplements to avoid LCPUFA shortfall. DHA and ARA have important regulatory roles, and such deficits may not only compromise growth and neurodevelopment, but they can also contribute to an increase in prematurity-related morbidities [7,8]. The use of lipid emulsions containing LCPUFAs serves to supply these fatty acids in premature neonates. Supplementation of enteral and parenteral feeding with ARA and DHA is used worldwide with safety. However, their doses and ratio remain to be defined. Fish oil-based emulsions are commonly used as source of DHA. Marine oils have very low ARA levels, and it is suggested that there could be possible negative effects on growth [9] due to the further decline in ARA along the early neonatal period [10]. However, in a recent retrospective study in preterm infants, it was concluded that fish oil did not negatively affect weight gain [11]. Sufficient ARA supply may be important for other reasons. This was put forward in a recent review, suggesting caution with formulas containing high DHA-to-ARA ratios, based on clinical and experimental data demonstrating reduced ARA content in erythrocytes and brain and worse cognitive outcomes [12]. The committee of the European Society for Pediatric Gastroenterology, Hepatology, and Nutrition proposed that the ARA:DHA ratio should be in the range of 1:1 or 2:1 [13].

LCPUFAs can also influence oxidative balance, but there is still controversy regarding their role, particularly during gestation and postnatal life. Both antioxidant and pro-oxidant effects were demonstrated using supplementations with *n*-3 LCPUFAs. DHA exhibits in vitro scavenging actions against ROS, limiting oxidative stress [14,15]. In addition, *n*-3 LCPUFA intake has beneficial effects against ROP, through protection of photoreceptors from oxidative stress [16], and supplementation during pregnancy and lactation decreased plasma hydroperoxides in newborns [17]. However, it was also observed that, under high oxidant stress conditions, LCPUFAs can be attacked by ROS and be oxidized into lipid peroxides, which are associated with oxidative damage and inflammation [18]. It was proposed that these dual effects of LCPUFAs may depend on their dose, and on the previous oxidative status of the subject, i.e., on the amount of antioxidant defenses [17,19]. This may be relevant in preterm neonates, who are endowed with an immature antioxidant defense system.

The majority of interventions assessing the effects of LCPUFA supplementation on oxidative balance used *n*-3 fatty acids. As indicated above, both ARA and DHA are needed for preterm infant postnatal development. We hypothesize that supplementation of premature neonates with a lipid emulsion based on ARA:DHA at a 2:1 ratio during the early perinatal period improves oxidative status. In this pilot study, in 21 preterm neonates, we compared the effects of this supplementation (ARA:DHA-S) with a control lipid supplement (medium-chain triglycerides, MCT-S), evaluating plasma *n*-3 and *n*-6 LCPUFAs, antioxidants, and oxidative damage biomarkers at birth, postnatal day 28, and 36 weeks of postmenstrual age (36 WPM). ARA:DHA-S improved oxidative status. Despite no change in systemic DHA values, a decrease in fatty acid precursors in the *n*-6 pathway relative to the *n*-3 pathways was found. These data support the potential benefits of ARA:DHA supplementation against oxidative stress and inflammation. 

## 2. Materials and Methods 

### 2.1. Study Population

A randomized, double-blind, pilot study was conducted in the neonatal intensive care unit (NICU) of La Paz University Hospital (HULP; Madrid, Spain). Twenty-one preterm infants were included in the study. The inclusion criteria were male and female preterm neonates, born ≤31 weeks + six days of gestational age, admitted to the NICU-HULP from January to May 2019. The exclusion criteria were congenital malformations or metabolic alterations, chromosomopathies, and newborns born to mothers taking prenatal DHA supplements.

The work was carried out in accordance with the code of ethics and complies with the Declaration of Helsinki, involving studies in humans, and it was approved by the HULP Research Ethics Committee (HULP-4287, protocol code HULP/NEO_DHA_01). Written informed consent was obtained from the mother or father prior to inclusion of each infant.

### 2.2. Supplementation 

The experimental lipid supplement was an oil derived from fungi and microalgae in the form of triglycerides, containing ARA (C20:4 *n*-6) and DHA (C22:6 *n*-3) in the proportion of 2:1 (ARA:DHA-S). This oil is available for infant formula products (Formulaid™ 2:1, DSM Nutritional Products, Switzerland). The control lipid supplement (MCT NM) was an oil enriched with medium-chain triglycerides (MCT-S). It was chosen as a control since it does not contain ARA, DHA, or their precursors. This lipid is used as a food supplement for special medical purposes (Cantabria Labs, Nutrición Médica SL, Spain). It was chosen as a control supplementation since it lacks LCPUFAs. The doses of both lipid supplements were matched to provide the same caloric energy. The LCPUFA content of both supplementations is shown in Table 1.

Group assignment was made after parents gave their written informed consent. Random allocation software was used with a 1:1 ratio [20]. The randomization was performed by a hospital staff who did not participate in the research study. A code was assigned to each intervention and sequentially numbered.

The lipid supplements were prepared with an aseptic technique using the hospital’s pharmacy service and the physical characteristics of the MCT-S and the ARA:DHA-S were blinded to all staff. The volume of ARA:DHA-S was calculated daily, based on the weight of the child, to deliver 40 mg/kg DHA and 80 mg/kg ARA per day, which is the estimated intrauterine accretion of LCPUFAs during the last trimester of pregnancy [21]. The energy provided by this dose was then calculated, and neonates on MCT-S received the volume of supplement providing the equivalent caloric intake. 

There was no intrauterine supplementation. After birth, the neonates received macronutrients through parenteral feeding with SMOFlipid^TM^, a glucose–saline solution, and Primene^TM^. Intravenous administration started at 2 g/kg/day, 2.5 g/kg/day, and 8 g/kg/day, with the goal of achieving 3, 12, and 3.5 g/kg/day, respectively. The babies were maintained on parenteral feeding until they tolerated full enteral feeding. Enteral feeding with human milk (breastmilk or donor milk) was given through an oro- or naso-gastric tube, and it was administered every 3 h.

Supplementation with ARA:DHA-S or MCT-S started the day the infant tolerated 30 mL/kg of enteral feeding and continued until 36 weeks postmenstrual age (WPA) or discharge, whichever occurred earlier. The supplements were administered in three aliquots, at 9:00 a.m., 6:00 p.m., and 12:00 a.m. Each dose was given by the NICU nurse as a bolus through the orogastric tube or by mouth immediately before human milk administration. Both supplementations were well tolerated, and no adverse events were reported.

### 2.3. Clinical and Anthropometrical Parameters 

At birth, the following parameters were abstracted from the medical record: maternal age (years), gestational age (weeks), Apgar scores at 1 and 5 min, Score for Neonatal Acute Physiology (SNAP)-II at birth [22], and neonatal sex. Anthropometric characteristics, weight (g), length (cm), and head circumference (cm) were measured at birth, at postnatal day 28, and at 36 WPA. The *z*-scores at birth were calculated according to the Fenton growth curves [23]. The intrauterine growth restriction was categorized as birth weight *z*-score lower than −1.3. Growth weight velocity was calculated according to an exponential model [24] and was expressed as g/kg/day, and the length and head circumference velocities were calculated according to a linear model [25] and were expressed as cm/day.

### 2.4. Blood Sample Extraction and Plasma Collection 

A venous blood sample was collected in Vacutainer^®^ tubes containing sodium citrate, at birth (before supplementation), at postnatal day 28, and at the end of supplementation (36 WPA or discharge, whichever occurred earlier). To avoid unnecessary punctures, the blood sample was obtained together with routine clinical tests. Plasma was obtained by centrifugation of the blood (2100× *g*, 15 min at 4 °C), within a maximum of 2 h of extraction. Thereafter, it was immediately aliquoted and stored at −80 °C.

### 2.5. Plasma Parameters of Oxidative Status 

*Total carbonyl groups*. Plasma protein carbonyls were assessed by the 2,4-dinitrophenylhydrazine-based assay, adapted by our group, as previously described [26]. The protein carbonyl concentration was determined using the extinction coefficient of 2,4-dinitrophenylhydrazine (ε = 22,000 M/cm). The absorbance was measured at 370 nm in a microplate reader (Synergy HTMultimode; BioTek). Data were expressed as nmol/mg of protein. Protein content was assessed by a Coomassie-blue-based microliter plate assay, according to the manufacturer’s instructions (Bio-Rad).

*Malondialdehyde (MDA).* The concentration of plasma MDA was measured by a spectrophotometric method detecting thiobarbituric acid (TBA) reactive substances, adapted by Reference [27]. Briefly, the plasma samples were incubated with trichloroacetic acid (TCA), ethylenediaminotetraacetic acid (EDTA), sodium dodecyl sulfate (SDS), and butylhydroxytoluene (BHT), followed by the addition of TBA, and they were boiled in a water bath at 100 °C for 30 min. After cooling, the mixture was centrifuged at 10,000× *g*, and the absorbance measured at 532 nm and compared with a standard curve of 1,1,3,3-tetrathoxypropane. The MDA was expressed as μmol/mL.

*Reduced glutathione (GSH).* Plasma GSH was assessed by a fluorometric method based on the reaction with *o*-phthalaldehyde [28], adapted to a microplate reader. Fluorescence was measured at 360 ± 40 nm excitation and 460 ± 40 nm emission wavelengths. GSH concentration in the samples was expressed as μmol/mg of protein.

*Total thiol groups.* Plasma thiols were assessed by a modification of the method based on 5,5′-dithiobis (2-nitrobenzoic acid) assay, adapted by us to a microplate reader, as previously described [29]. The absorbance was measured at 412 nm, and thiol content was expressed as nM GSH/mg of protein.

*Catalase activity.* Catalase activity was assessed by Amplex Red catalase assay (Enz-Chek Myeloperoxidase Assay Kit with Amplex Ultra Red reagent; Invitrogen). Catalase activity was expressed as U/mg of protein.

*Calculation of oxidative status scores.* The antioxidant score (Antiox-S) was calculated for each subject taking into account the abovementioned antioxidant parameters in plasma (GSH, total thiol groups, and catalase activity) using the statistical methodology previously described [29]. Similarly, the score related to oxidative damage (Proxy-S) was also calculated from MDA and total carbonyl groups. The Antiox-S:Proxy-S index was also calculated.

*Change in the oxidative status levels.* The change (Δ) in oxidative status parameters at 36 WPA was calculated by subtracting, for each individual, the levels at birth from the levels at the end of supplementation. 

### 2.6. Plasma LCPUFAs 

Lipid extraction was carried out using the Folch method [30], followed by methylation and quantification by gas chromatography–mass spectroscopy (GC–MS). Briefly, 100 μL of plasma was added to 400 µL of phosphate-buffered saline solution and 30 μg of heptadecanoic acid (17:0) to serve as the internal standard. Fatty acids from plasma preparations were isolated by the addition of chloroform–methanol (2:1 *v*/*v*). The lipid samples were vortexed, sonicated, and centrifuged at 400× *g* for 5 min at 4 °C. The organic phase was removed, and fatty acids were methylated and quantified with a Hewlett-Packard Series II 5890 chromatograph (GMI, Ramsey, MN; USA) coupled to a HP-5971 mass spectrometer (LabX, Midland, ON; Canada), equipped with a Super-Cowax SP–10 capillary column (fused silica, inner diameter (i.d.) = 0.10 μm, 15 m × 0.10 mm). The carrier gas flows through the central aperture and is unrestricted throughout the length of the column. This column is based on one of the most widely used polar phases, Carbowax 20 M, suitable for analyses of fatty acid methyl esters (FAMEs). The column temperature was held at 100 °C during the injection. Then, it was raised at the rate of 30 °C per minute to 190 °C, held at 190 °C for 8 min, and then raised again to 230 °C at the rate of 30 °C per minute. The column temperature was maintained at 230 °C for an additional 3 min, and it was cooled to 100 °C for the next analysis.

For the fatty-acid quantification, peak identification was based upon comparison of both retention time and mass spectra of the unknown peak to that of known standards within the GC–MS database library. FAME mass was determined by comparing areas of unknown FAMEs to that of a fixed concentration of 17:0 internal standard. The levels of fatty acids were reported in nmol.%. The saturated and monounsaturated fatty acids were reported as a sum of all of them. The *n*-6:*n*-3 LCPUFAs, linoleic acid (LA; 18:2 *n*-6):DHA, and ARA:DHA ratios were calculated.

*Change in the LCPUFAs levels.* The changes (Δ) in the level of each LCPUFA at 36 WPA were calculated by subtracting, for each individual, their level at birth from the level at 36 WPA.

### 2.7. Statistical Analysis

Statistical analysis was performed with SPSS Statistic version 25.0 (IBM Company, Armonk, NY; USA). Some variables in this study did not follow a normal distribution. Since the sample size was small (21 infants), we considered it more appropriate to use nonparametric methods. Therefore, quantitative variables were expressed as medians and interquartile ranges (IQRs), and qualitative variables were expressed as a relative frequency and sample size (*n*). The U Mann–Whitney test was used to analyze differences in the variables according to diet supplementation, and Spearman’s rho correlations (*r*) were used to determine the association between variables. Statistical significance was defined as *p* < 0.05.

## 3. Results

### 3.1. Neonatal Cohort and Baseline Characteristics

Twenty-one subjects, eight males and 13 females, were recruited. Baseline characteristics according to diet supplementation are depicted in Table 2. No statistical differences were detected at birth between MCT-S and ARA:DHA-S groups with the exception of birth length *z*-score. The infants in the ARA:DHA-S group were longer than the infants in the MCT-S group.

To compare the effects of MCT-S and ARA:DHA-S, we analyzed the changes in growth, and plasma levels of LCPUFAs and oxidative status from birth to 28 days of postnatal life or to the end of the supplementation period (36 WPA or discharge).

### 3.2. Diet Supplementation and Plasma LCPUFA Profiles

At birth, LCPUFAs did not show statistical differences between male and female infants (Appendix A).

At 28 days and 36 weeks postmenstrual age, the levels of linoleic and dihomo-γ-linolenic acids were significantly lower, and α-linolenic acid level was significantly higher in the ARA:DHA-S group than the MCT-S group. The ratios between *n*-6:*n*-3 LCPUFAs and ARA:DHA were similar between diet supplementation groups. However, the LA:DHA ratio was significantly lower during diet supplementation in the ARA:DHA-S group compared to the MCT-S group (Table 3).

To compare the effect of the supplementations on the global pattern of the LCPUFAs, we also calculated for each individual the increment (Δ) as the subtraction between the final level at 36 WPA and the first level at birth. 

In the ARA:DHA-S group, the Δ in arachidonic acid was significantly higher, but the linoleic acid, the eicosapentaenoic acid, and the ratio of LA:DHA were significantly lower compared to the MCT-S group (Figure 1).

The Δ saturated fatty acids, monounsaturated fatty acids, and n-6:n-3 LCPUFA ratio did not show statistically significant differences between diet supplementation groups. 

### 3.3. Diet Supplementation and Plasma Oxidative Status

At birth, we did not detect statistically significant differences between male and female infants in plasma antioxidants or biomarkers of oxidative damage (Appendix A).

At the beginning of supplementation, none of the oxidative parameters showed significant differences between groups. At 28 days of postnatal life, the lipid peroxidation levels (malondialdehyde) were significantly higher in ARA:DHA-S compared to MCT-S. This pattern was also observed at 36 weeks postmenstrual age. At this time point, the score of oxidative damage (Proxy-S) was also significantly higher in ARA:DHA-S compared to MCT-S. Additionally, at the end of supplementation, the levels of catalase activity and reduced glutathione, and the global antioxidant score (Antiox-S) were significantly higher in ARA:DHA-S than in MCT-S (Table 4).

We evaluated the change (Δ) in oxidative status parameters from birth until the end of supplementation. By 36 WPA, catalase activity, total thiol groups, and Antiox-S were all significantly higher in neonates receiving ARA:DHA-S in comparison to those receiving MCT-S (Figure 2). Regarding plasma biomarkers of oxidative damage, we did not detect statistically significant differences between infants supplemented with ARA:DHA or MCT at the end of the supplementation period (Figure 2). We did not detect significant differences in the Antiox-S:Proxy-S index between groups.

In the ARA:DHA-S group, we found a positive correlation between the change in Antiox-S and the change in Proxy-S from birth to the end of supplementation (*r* = 0.747, *p* = 0.003).

### 3.4. Effect of Diet Supplementation on Neonatal Growth

All anthropometric parameters at birth were larger in male compared to female infants (Table 5). In both sexes, statistically significant positive correlations were found between gestational age and anthropometric parameters at birth (weight: *r* = 0.563, *p* = 0.008; length: *r* = 0.448, *p* = 0.047; head circumference: *r* = 0.719, *p* = 0.001). Statistically significant negative correlations were found between gestational age and oxidative score at birth (Proxy-S: *r* = −0.480, *p* = 0.044).

Increases in body weight, length, and head circumference were not statistically significant different between MCT-S and ARA:DHA-S groups, either by day 28 of postnatal life (data not shown) or by the end of treatment. Similarly, no significant differences were detected in *z*-scores or growth velocities between groups (Appendix A).

## 4. Discussion

Preterm neonates are deficient in ARA and DHA, and they have immature antioxidant systems. This makes them more vulnerable to perinatal morbidities, such as ROP, NEC, BPD, and ductus arteriosus, which are associated with oxidative damage and inflammation. Therefore, the use of nutritional strategies with antioxidant capacity may reduce the incidence or severity of these pathologies and improve infant development. This pilot study demonstrates that supplementation of premature neonates in the early neonatal period with a lipid emulsion based on ARA:DHA in a 2:1 ratio is able to improve antioxidant status without compromising growth. The change in plasma antioxidants was associated with reduced linoleic acid levels and LA:DHA ratio. The improvement in antioxidants and the modifications in the profile of *n*-3 and *n*-6 fatty acid precursors may have a positive impact on oxidative stress and inflammation.

Premature infants are unable to overcome the decline in blood DHA or ARA levels that occurs after birth, and they require supplementation with these LCPUFAs. However, there is still controversy on the best ratios of ARA:DHA. Based on clinical and experimental studies, it was suggested that ARA should be equal to or higher than DHA [12,13]. In the present study, we chose to use a 2:1 ARA:DHA ratio for three reasons. Firstly, this ratio was shown to represent accretion in gestation [6]. Secondly, it is closer to human milk content, which is higher in ARA [31]. Thirdly, we have previous evidence, from a randomized clinical trial in very preterm neonates, that infants who received formula with an ARA:DHA ratio of 2:1 had higher blood levels of essential fatty acids during the first year of life, and better psychomotor development, compared to those who received 1:1 ratio [32].

### 4.1. Effect of Supplementation on Plasma LCPUFAs

In neonates receiving ARA:DHA-S, we did not detect an increase in plasma DHA compared to those receiving MCT-S. However, an increase in ARA was observed, together with a decrease in linoleic acid, a precursor of the *n*-6 pathway. We suggest that this reduction could be beneficial in terms of inflammation. The biological activity of PUFAs seems to be mediated, at least in part, via oxylipins, synthesized by enzymatic systems or through ROS mediate oxidation. Linoleic and α-linolenic acids are one of the sources of oxylipins. Oxylipins derived from linoleic acid are mostly pro-inflammatory, particularly under pro-oxidant conditions [33,34], and they are associated with respiratory distress syndrome in rats [35] and oxidative stress in vitro [33].

The lack of change in plasma ARA:DHA in Formulaid^TM^-supplemented infants, despite the reduction in their precursors, may indicate that the exogenous fatty acids from the supplement are being absorbed and the endogenous synthesis is reduced. LCPUFA of *n*-6 and *n*-3 series are known to compete in the metabolic pathways and for the incorporation into phospholipids. A study in primates showed greater eicosapentaenoic acid incorporation into plasma and erythrocyte lipids when the level of dietary linoleic acid was reduced [36]. Studies in humans also demonstrated that diets with *n*-6 LCPUFAs hinder the incorporation of *n*-3 LCPUFAs into plasma and tissue lipids [37,38].

### 4.2. Effect of Supplementation on Oxidative Status 

LCPUFAs of the *n*-3 and *n*-6 series were proposed to reduce the impact of hypoxia and oxidative damage [39]. However, there is still controversy regarding the effect of these fatty acids on oxidative stress, since, in the presence of high ROS concentrations, they may undergo oxidation producing peroxides [18]. The present study demonstrates that neonates receiving ARA:DHA 2:1 supplementation increased plasma levels of some endogenous antioxidants and the Antiox-S score. We previously used this parameter in humans with several pathologies associated with oxidative stress, namely, pregnancy complication [29], venous insufficiency [26], and hypertension [40]. This score is useful to assess global antioxidant status when individual parameters are highly variable.

The improvement of antioxidant status of preterm neonates receiving LCPUFA supplementation was only detected at the end of the supplementation period. A tendency toward an increase in Antiox-S was observed at day 28, but did not reach statistical significance (*p* = 0.069). In a study with a similar intervention (ARA:DHA 1.6:1), supplementation for seven days did not modify total antioxidant capacity in preterm neonates [41]. These data suggest that long periods of supplementation might be needed to produce an effect. In addition to the shorter supplementation period of the study by Siahanidou et al., another difference was the larger gestational age of the population they analyzed, compared to ours. It is, therefore, possible that ARA:DHA supplementation may exert larger effects in infants with more immature antioxidant systems. We found a negative correlation between gestational age and Proxy-S at birth, suggesting that oxidative damage is higher in children with a lower level of maturation. Higher levels of oxidative damage biomarkers in children with lower gestational age may also be related to the use of positive pressure ventilation and intubation, interventions which are known to increase the levels of ROS [3].

Although we found an improvement in antioxidant status with ARA:DHA-S, we did not find a reduction in oxidative damage biomarkers for proteins and lipids or in the Proxy-S. This is in contrast with a previous study using SMOFLipid supplementation in preterm infants, which showed a reduction in F2-isoprstanes in urine [42]. It is possible that this is due to differences in the changes in fatty acids (we detect a change in ARA, but not in DHA or eicosapentaenoic acids), the methodology used, or the biomarker analyzed. However, we consider a second explanation. We found a positive correlation between Antiox-S and Proxy-S scores at the end of the intervention period with ARA:DHA-S. We suggest that supplementation with LCPUFA may be inducing mild oxidative stress related to the formation of peroxides. It was proposed that a mild pro-oxidative status may serve as stimulus for activation of endogenous antioxidants, through a hormetic mechanism [43]. This was previously demonstrated in the context of physical exercise, where moderate exercise and a mild pro-oxidative state induce an increase in endogenous antioxidant defenses, while high-intensity exercise leads to oxidative damage [44]. Therefore, we propose that the ARA:DHA supplementation may exert a similar effect, stimulating antioxidant defenses in premature neonates. 

### 4.3. Effect of LCPUFA on Infant Growth 

The LCPUFA intervention did not have an effect on growth rate. Similar data were described with SMOFLipid, an emulsion based on fish oil including ARA and DHA [42] or with a combination of ARA:DHA similar to the one used in this study [41]. Although we did not detect an improvement in growth, it is possible that this supplementation might reduce perinatal morbidities or improve neurodevelopment, an aspect which should be evaluated in a larger cohort.

### 4.4. Influence of Sex 

Female fetuses and infants seem to be better adapted to adverse intrauterine and perinatal environments. Female fetuses have an advanced level of maturation compared to males, and female placentas exhibit a more adaptive response to stress factors such as undernutrition [45]. Prematurity is more prevalent in male infants [46], and they are also at higher risk of perinatal pathologies compared to female preterm neonates [47,48]. Based on these facts, it could be hypothesized that preterm female neonates might have a better acquisition of LCPUFAs and maturation of their antioxidant systems. However, we did not detect differences between sexes in the plasma LCPUFAs or antioxidants at birth. Lack of differences between male and female infants in antioxidant status may be related to the fact that we analyzed plasma. A recent systematic review evidenced that sex-specific differences in antioxidants are observed mostly in tissues (i.e., the intracellular compartment), rather than in biological fluids [49]. It is important to consider the small sample size, which is a limitation of the present study, making it difficult to draw conclusions on the influence of sex on oxidative status or LCPUFAs. Given the importance of sex as a biological variable for clinical practice, further research in a larger cohort of preterm male and female infants would aid to confirm or discard its influence on the studied variables.

## 5. Conclusions

DHA and ARA diet supplementation improved oxidative status despite no change in systemic values in DHA. However, we did show a reduction in the precursor fatty acids of the n-6 LCPUFA pathway (linoleic acid) suggesting active signaling from these fatty acids in the oxidative status, probably via inflammatory mediators. Whether this change in oxidative status with supplementation reduces morbidities needs to be evaluated in a large clinical trial.

## Figures and Tables

**Figure 1 nutrients-12-00122-f001:**
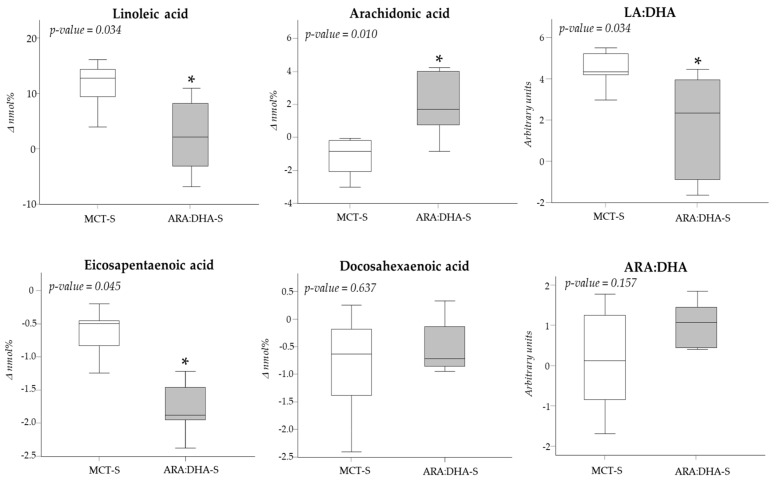
Changes in plasma long-chain polyunsaturated fatty acids (LCPUFAs) from birth to 36 week of postmenstrual age according to medium-chain triglyceride (MCT) (*n* = 8) or arachidonic acid/docosahexaenoic acid (ARA:DHA) (*n* = 6) supplementation. Linoleic acid (LA), docosahexaenoic acid (DHA), arachidonic acid (ARA). Data show medians and interquartile ranges (IQR). * *p* < 0.05; Mann–Whitney U test.

**Figure 2 nutrients-12-00122-f002:**
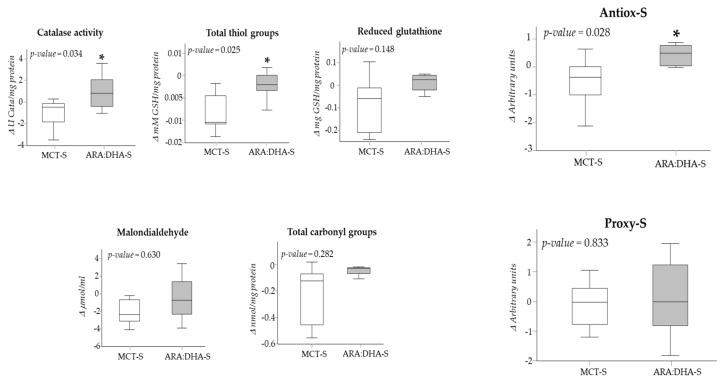
Changes in plasma antioxidants, the global antioxidant score (Antiox-S), biomarkers of oxidative damage, and the global score of oxidative damage (Proxy-S) from birth to 36 weeks of postmenstrual age according to MCT (*n* = 8) or ARA:DHA (*n* = 6) supplementation. Data show medians and IQR. * *p* < 0.05; Mann–Whitney *U* test.

**Table 1 nutrients-12-00122-t001:** Fatty-acid profile used for diet supplementation in the infant cohort. MCT—medium-chain triglycerides.

Figure 12.	Formulaid^TM^	MCT NM
Lauric Acid (C12:0)	1.3	-
Myristic Acid (C14:0)	3.1	0.1
Palmitic Acid (C16:0)	14.2	16.3
Palmitoleic Acid (C16:1 *n*-7)	0.5	0.2
Stearic Acid (C18:0)	4.5	2.7
Oleic Acid (C18:1 *n*-9)	20.8	20.4
Linoleic Acid (C18:2 *n*-6)	27.5	53.5
Linolenic Acid (C18:3 *n*-6)	1.0	-
Alpha Linoleic Acid (C18:3 *n*-3)	3.0	4.6
Arachidic Acid (C20:0)	0.4	0.3
Gondoic Acid (C20:1 *n*-9)	0.2	0.3
Eicosadienoic Acid (C20:2 *n*-6)	0.2	-
Eicosatrienoic Acid (C20:3 *n*-6)	0.9	-
Arachidonic Acid (C20:4 *n*-6)	13.5	-
Eicosapentaenoic Acid (C20:5 *n*-3)	0.1	-
Behenic Acid (C22:0)	0.5	-
Docosahexaenoic Acid (C22:6 *n*-3)	6.3	-
Lignoceric Acid (C24:0)	0.4	-

Formulaid^TM^ was used in the arachidonic acid/docosahexaenoic acid (ARA:DHA-S) infant group; MCT NM was used in MCT-S infant group.

**Table 2 nutrients-12-00122-t002:** Cohort baseline characteristics.

	MCT-S (*n* = 12)	ARA:DHA-S (*n* =1 0)	*p*-Value
Maternal age (years)	34.5 (9.0)	31.0 (3.0)	0.447
Gestational age (weeks)	28.9 (2.6)	27.9 (1.4)	0.382
Apgar at 1 min	7.0 (1.3)	7.0 (1.0)	0.602
Apgar at 5 min	8.0 (1.0)	8.0 (2.0)	0.549
Score Neonatal Acute Physiology-II	24.0 (15.3)	23.0 (5.0)	0.862
Sex (female)	50.0% (6)	85.7% (7)	0.112
Birth weight (g)	856.0 (314.5)	1075.0 (381.0)	0.241
Birth weight *z*-score	−0.12 (1.5)	0.4 (0.8)	0.102
Intrauterine growth restriction	16.7% (2)	0% (0)	0.198
Birth length (cm)	34.0 (4.4)	35.5 (4.3)	0.844
Birth length *z*-score	−0.63 (2.2)	0.45 (1.0) *	0.020
Birth head circumference (cm)	25.0 (4.3)	25.5 (3.2)	0.262
Birth head circumference *z*-score	−0.31 (1.2)	0.98 (1.9)	0.080

Data show medians (IQR) in quantitative variables or relative frequency (*n*). * *p* < 0.05; Mann–Whitney **U** test.

**Table 3 nutrients-12-00122-t003:** Levels of long-chain polyunsaturated fatty acids (LCPUFAs) and ratios at different points according to supplementation.

	MCT-S (*n* = 10)	ARA:DHA-S (*n* = 12)	*p*-Value
**Linoleic acid** (nmol %)			
At birth	11.10 (5.0)	15.52 (5.36)	0.063
At 28 DPL	19.38 (0.57)	15.74 (3.78) *	0.005
At 36 WPA	22.80 (1.41)	18.54 (1.68) *	0.018
**Dihomo-****γ-linoleni****c acid** (nmol %)			
At birth	1.60 (0.41)	1.40 (0.69)	0.518
At 28 DPL	2.34 (0.48)	1.96 (0.37) *	0.040
At 36 WPA	2.32 (0.58)	1.68 (0.38) *	0.018
**Arachidonic acid** (nmol %)			
At birth	8.31 (1.54)	7.45 (3.45)	0.470
At 28 DPL	7.91 (1.48)	8.29 (2.47)	0.329
At 36 WPA	7.50 (1.27)	9.27 (2.34)	0.050
**α-Linolenic acid** (nmol %)			
At birth	0.23 (0.19)	0.78 (0.49)	0.121
At 28 DPL	0.16 (0.09)	0.50 (0.35) *	0.006
At 36 WPA	0.22 (0.10)	0.52 (0.33) *	0.018
**Eicosapentaenoic acid** (nmol %)			
At birth	0.78 (0.65)	1.63 (0.57) *	0.037
At 28 DPL	0.55 (0.80)	0.62 (0.90)	0.491
At 36 WPA	0.28 (0.31)	0.44 (0.24)	0.289
**Docosahexaenoic acid** (nmol %)			
At birth	3.07 (1.40)	3.08 (0.91)	0.909
At 28 DPL	2.79 (0.62)	3.08 (0.58)	0.770
At 36 WPA	2.61 (1.17)	2.83 (0.44)	0.346
**Saturated fatty acids** (nmol %)			
At birth	32.41 (3.29)	31.97 (1.95)	0.970
At 28 DPL	32.89 (2.47)	35.40 (0.66)	0.064
At 36 WPA	33.71 (1.91)	35.70 (2.07)	0.077
**Monounsaturated fatty acids** (nmol %)			
At birth	38.72 (3.15)	35.85 (3.33) *	0.007
At 28 DPL	30.55 (2.93)	30.75 (3.70)	0.380
At 36 WPA	29.21 (2.46)	30.26 (3.40)	0.480
***n*****-6:*n*-3 LCPUFAs** (arbitrary units)			
At birth	3.88 (1.46)	3.91 (1.10)	0.970
At 28 DPL	6.75 (2.45)	5.16 (1.38)	0.097
At 36 WPA	8.17 (1.78)	7.48 (1.62)	0.389
**Linoleic acid:docosahexaenoic acid** (arbitrary units)			
At birth	4.01 (0.69)	5.04 (1.17) *	0.020
At 28 DPL	6.57 (1.06)	4.67 (0.84) *	0.006
At 36 WPA	8.21 (2.65)	6.26 (2.35) *	0.045
**Arachidonic acid:docosahexaenoic acid** (arbitrary units)			
At birth	2.65 (1.56)	2.52 (1.35)	0.790
At 28 DPL	2.74 (1.15)	2.59 (0.84)	0.558
At 36 WPA	2.95 (0.77)	3.41 (0.61)	0.346

Data show medians (IQR). Days of postnatal life (DPL), weeks of postmenstrual age (WPA). *p*-values compare ARA:DHA-S with MCT-S at each point. Supplements were given in three aliquots, at 9:00 a.m., 6:00 p.m., and 12:00 a.m. * *p* < 0.05; Mann–Whitney *U* test.

**Table 4 nutrients-12-00122-t004:** Plasma oxidative status at different time points according to supplementation.

	MCT-S (*n* = 10)	ARA:DHA-S (*n* = 12)	*p*-Value
**Catalase activity** (*U* catalase/mg protein)			
At birth	3.37 (3.16)	3.86 (1.45)	0.644
At 28 DPL	3.04 (1.63)	4.61 (0.89)	0.085
At 36 WPA	2.86 (0.94)	4.74 (0.40) *	0.025
**Total thiol groups** (mM GSH/mg protein)			
At birth	0.013 (0.011)	0.011 (0.002)	0.337
At 28 DPL	0.007 (0.002)	0.010 (0.005)	0.134
At 36 WPA	0.007 (0.002)	0.009 (0.003)	0.113
**Reduced glutathione** (mg GSH/mg protein)			
At birth	0.15 (0.22)	0.20 (0.08)	0.915
At 28 DPL	0.09 (0.26)	0.25 (0.12)	0.135
At 36 WPA	0.07 (0.12)	0.26 (0.19) *	0.021
**Antiox-S** (arbitrary units)			
At birth	−0.10 (1.64)	0.03 (0.52)	0.702
At 28 DPL	−0.63 (1.11)	0.36 (0.35)	0.069
At 36 WPA	−0.67 (0.96)	0.55 (0.35) *	0.045
**Malondialdehyde** (μmol/ml)			
At birth	4.36 (1.44)	6.46 (2.27)	0.118
At 28 DPL	2.79 (0.68)	4.63 (1.29) *	0.005
At 36 WPA	2.41 (0.90)	4.81 (3.70) *	0.008
**Total carbonyl groups** (nmol/mg protein)			
At birth	0.29 (0.36)	0.25 (0.20)	0.670
At 28 DPL	0.25 (0.29)	0.42 (0.40)	0.342
At 36 WPA	0.12 (0.14)	0.22 (0.27)	0.556
**Proxy-S** (arbitrary units)			
At birth	−0.01 (1.24)	0.38 (0.72)	0.382
At 28 DPL	−0.29 (0.43)	0.39 (0.93)	0.052
At 36 WPA	−0.27 (0.64)	0.27 (1.15) *	0.010
**Antiox-S:Proxy-S** (arbitrary units)			
At birth	1.19 (2.18)	0.07 (1.22)	0.148
At 28 DPL	0.93 (4.10)	0.43 (0.62)	0.995
At 36 WPA	0.80 (1.38)	0.79 (8.37)	0.456

Data show medians (IQR). Days of postnatal life (DPL), weeks of postmenstrual age (WPA). *p*-values compare ARA:DHA-S with MCT-S at each point. Supplements were given in three aliquots, at 9:00 a.m., 6:00 p.m., and 12:00 a.m. * *p* < 0.05; Mann–Whitney *U* test.

**Table 5 nutrients-12-00122-t005:** Cohort baseline characteristics between infant sexes

	Male (*n* = 8)	Female (*n* = 13)	*p*-Value
Maternal age (years)	33.0 (6.0)	31.0 (10.0)	0.840
Gestational age (weeks)	27.0 (4.0)	28.0 (2.0)	0.916
Apgar at 1 min	7.0 (2.5)	7.0 (2.0)	0.972
Apgar at 5 min	8.5 (2.0)	8.0 (2.0)	0.778
Score of Neonatal Acute Physiology-II	23.5 (10.5)	23.0 (6.0)	0.743
Weight (g)	1192.5 (472.5)	856 (200)	0.030
Weight *z*-score	0.44 (0.5)	−0,17 (0.6)	0.161
Intrauterine growth restriction	12.5% (1)	7.7% (1)	0.716
Length (cm)	38 (4.0)	34 (1.5)	0.030
Length *z*-score	0.40 (0.5)	−0.50 (1.3)	0.196
Head circumference (cm)	26.9 (3.5)	24.4 (3.5)	0.053
Head circumference *z*-score	1.16 (1.4)	−0.58 (0.5)	0.020

Data show medians (IQR) in quantitative variables or relative frequency (*n*) in qualitative variables. Mann–Whitney *U* test.

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
