# Peer review of "Plasma Oxidative Status in Preterm Infants Receiving LCPUFA Supplementation: A Pilot Study"

_nutrients, 2020, doi:10.3390/nu12010122_

Round 1
Reviewer 1 Report
The manuscript numbered 677496 deals with the oxidative status of plasma of preterm infants supplemented with the long-chained polyunsaturated fatty acids (LCPUFA). In general paper is well written, interesting and may be of interest for scientific audience. But several issues listed below still need improvements. Manuscript could be considered for publication in Nutrients but after minor revision.
Line 67: ‘…oxidative status of preterm infants during…” Lines 70 - 74: What was the total number of neonates included in this study? Please specify an exact ‘n’ value. Line 80: Does oil used in present study was commercially available? Line 83: Was this control oil also derived from fungi and microalgae? Lines 80 – 84: Detailed profile of FA of oils used for supplementation should be given in the manuscript. Lines 80 – 84: Please specify an exact dose of lipid supplement administrated to the neonates during experimental period. Line 151: Folch method does not serve for derivatization of FA but only for fat extraction. This should be corrected. Line 153: Detailed parameters of chromatographic column used for GC-MS analysis should be specified. Line 164: When the variables does not have a normal distribution non-parametric statistical test should be used for analysis. What was an exact sample size? Line 166: What are ‘Spearman’s rho correlations’? Line 172: Are there any changes according to the gender of neonates? Line 285: The ‘very preterm’ expression should be specified. How many days/month before the planned delivery is ‘very’?
Author Response
We would like to thank the reviewer for the comments and suggestions. Please find the point by point answers and the changes in the new version of the manuscript (Please see the attachment). We have now included a table with the fatty acid content of the supplements. The manuscript has also been edited to improve English language and style.

Reviewer 2 Report
This article is addressing the supplementation of pre-term infants by giving a different ratio normally given to this population. The authors do a great job in being forthcoming with their results. The treatment is then monitored for weeks to detect any differences in plasma FA composition, oxidative status, as well as infant characteristics. This article is of interest, however clarity/organization of writing and formation of conclusion based on data need to be addressed. Details are sometimes lacking and don't hesitate to repeat results as you discuss them.
1-make sure to spell out all acronyms (eg. LCPUFA in abstract).
2- Introduction is lacking enough background. My edits are detailed below. Refer to the discussion section, which should echo the introduction and vice versa.
(line 56-57) You mention that there is still controversy on the best ratio of n-6:n-3 LCPUFAs, please elaborate on this controversy. What are the pros and cons on each different ratios currently and previously used. You do this a bit in the discussion section (see comment regarding lines 294-304). It just needs to be a bit more so the audience has an idea of what is done currently and what the different opinions are regarding these ratios and their use in supplementation.
(line 294-304) This paragraph should be added to the introduction instead of being in the discussion section. This paragraph elaborates a bit LCPUFAs but does still need a bit more elaboration on ratios of DHA to ARA (for fish oil and marine oil etc...
(line 58, 59) Elaborate on oxidative status. You mentioned that it depends on the context... what context? give us details.
(line 60) In vitro needs to be italicized. Make sure to do this each time...
(lines 65-67) The last paragraph of the introduction should include the hypothesis, a sentence on method (on supplement treatment and analysis of lipid composition, ROS, …), a sentence on result/conclusion. This is a short summary of your study for the layman. In the 2nd paragraph of the discussion section (and in line 378-379) you write a summary of the study that should also be included here in the end of the introduction.There should be a hypothesis in the introduction, for example:
We hypothesize that supplementation of premature neonates with lipid emulsion based on ARA:DHA at 2:1 ratio improves oxidative status within [add time frame in weeks].
In the discussion section, mention if you support or reject this hypothesis (lines 280-282).
3- Methods: just a comment below.
Section 2.2 supplementation: (1)This is not clear: what was given intrauterine and how often? (2) what was given postpartum/after birth, how frequently, when during the day and/or when relative to feedings.
4- Results: Some explanations were lacking. My comments are detailed below.
(line 172) write out 21 - twenty on.
(line 174) Why is MCT-S a good control to use. Explain here or in methods.
Table 1: Why are you showing Median instead of Averages? it would seem that averages would be appropriate.
Section 3.2: Be sure to mention that you are collecting blood samples in the text. It is worth reminding the audience.
(line 197-198) It may be worth showing these data in supplement if space allows.
Figure 1: Where these changes measured by comparing group averages (which can be misleading)? or through individual changes, which were then averaged? Meaning each individual was measured at birth and week 36 to reveal changes, then each individual changes was averaged for the graph. The latter is more appropriate since you can see the number of individuals that are changing and how, instead of general changes that may not be reflective of individual changes.
(line 216) rewrite: No oxidative parameters showed significant differences…
(line 218) I believe there is a typo. instead of postmentrual, postpartum or postbirth.
Table 3: mention here again the frequency of the supplements.
(line 226-227) rerwrite: starting at birth continued until the end of supplementation.
(line 230) statistically significant differences instead of statistical difference.
5- Discussion: the writing was at times awkward, too casual and grammatically incorrect. I have detailed my comments below. I also commented on some of the conclusions you make, which you can fix through rewording appropriately.
(line 280-282) refer to earlier comment under introduction.
(line 285-287) Be clear on what part of your sentence you are referencing a study and which part you are presenting in this article. It is not clear stated as is.
(line 288) Remover "Besides" and change "show" to "shown"
(line 292) Change "compromise" to "compromising" and remove "to"
(line 294-304) refer to earlier comment. This paragraph should be in the introduction.
(line 316) Change: "show" to "shown", "whose" to "those", "level" to "levels"
(line 317-319) rephrase. These sentences are not clear and not structured well. I'm not clear on what you mean here.
(line 324) what ratio is ARA:DHA here?
(line 329-330) Remove "by the fact". Also since it was not shown to be statistically significant, it cannot be said that the increase was gradual based on the trend. Remove: from ", as shown" to "(data not reported), which." only mention that there was significant improvement by day 28.
(line 333-334) you mention that this is "another difference with Siahanidou et al (also italicize 'et al'), however it is not clear what the first difference is?
(line 337) Change: "probably" to "likely" (more based on numbers), "to by the use of interventions". Also interventions in some fields means that these are what the authors are doing to carry out their study. I would replace this with a synonym.
(line 340) change "despite the fact that" to "although". This is casual and inaccurate when not all are facts.
(line 348-349) "We expected the opposite trend..." Remind the audience what you found here so we know to compare with what you expected.
(line 353) remove "it"
(line360-361) Change "despite the fact" to "although"
(line 362) change "deserves to" to "should" (sounds less awkward)
(line 364) "Females" Is this referring to the infants or mothers? clarify the ‘Female/male you are referring to - do this throughout the paper.
(line 370-371) reword: reword: one of our limitations is the small sample size, which makes it difficult to draw conclusions on the influence of sex.
(line 371-372) Remove "besides" and mentioning "lack of difference" is inappropriate since you just said (in the previous sentence) that no conclusion could be drawn: remove sentence. It cannot be said that there is no difference when no conclusion can be formed (no difference is a conclusion).
(line 378-379) keep this sentence and also add it to the last paragraph of the introduction.
Author Response
Thank you for your comments and suggestions. Please find the point by point answers to your questions (Please see the attachment), and the changes in the new version of the manuscript. We have now included 2 Tables with the fatty acid content of the supplements (Table 1) and analysis of LCPUFAs according to sex (Supplemental Table 1). The manuscript has also been edited to improve English language and style.
